# A Multi-Resolution Approach to Point Cloud Registration without Control Points

Eleanor A. Bash [1,2,*], Lakin Wecker [3], Mir Mustafizur Rahman [2], Christine F. Dow [2], Greg McDermid [2], Faramarz F. Samavati [3], Ken Whitehead [4], Brian J. Moorman [2], Dorota Medrzycka [5] and Luke Copland [5]

1 Department of Geography and Environmental Management, University of Waterloo, Waterloo, ON N2L 3G1, Canada
2 Department of Geography, University of Calgary, Calgary, AB T2N 1N4, Canada
3 Department of Computer Science, University of Calgary, Calgary, AB T2N 1N4, Canada
4 Centre for Innovation and Research in Unmanned Systems, Southern Alberta Institute of Technology, Calgary, AB T2M 0L4, Canada
5 Department of Geography, Environment and Geomatics, University of Ottawa, Ottawa, ON K1N 6N5, Canada
* Correspondence: eleanor.bash@ucalgary.ca

**Abstract:** Terrestrial photographic imagery combined with structure-from-motion (SfM) provides a relatively easy-to-implement method for monitoring environmental systems, even in remote and rough terrain. However, the collection of in-situ positioning data and the identification of control points required for georeferencing in SfM processing is the primary roadblock to using SfM in difficult-to-access locations; it is also the primary bottleneck for using SfM in a time series. We describe a novel, computationally efficient, and semi-automated approach for georeferencing unreferenced point clouds (UPC) derived from terrestrial overlapping photos to a reference dataset (e.g., DEM or aerial point cloud; hereafter RPC) in order to address this problem. The approach utilizes a Discrete Global Grid System (DGGS), which allows us to capitalize on easily collected rough information about camera deployment to coarsely register the UPC using the RPC. The DGGS also provides a hierarchical set of grids which supports a hierarchical modified iterative closest point algorithm with natural correspondence between the UPC and RPC. The approach requires minimal interaction in a user-friendly interface, while allowing for user adjustment of parameters and inspection of results. We illustrate the approach with two case studies: a close-range (<1 km) vertical glacier calving front reconstructed from two cameras at Fountain Glacier, Nunavut and a long-range (>3 km) scene of relatively flat glacier ice reconstructed from four cameras overlooking Nàlùdäy (Lowell Glacier), Yukon, Canada. We assessed the accuracy of the georeferencing by comparing the UPC to the RPC, as well as surveyed control points; the consistency of the registration was assessed using the difference between successive registered surfaces in the time series. The accuracy of the registration is roughly equal to the ground sampling distance and is consistent across time steps. These results demonstrate the promise of the approach for easy-to-implement georeferencing of point clouds from terrestrial imagery with acceptable accuracy, opening the door for new possibilities in remote monitoring for change-detection, such as monitoring calving rates, glacier surges, or other seasonal changes at remote field locations.

**Keywords:** photogrammetry; structure-from-motion; Discrete Global Grid System; DGGS; change detection; point cloud registration

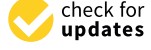



## 1. Introduction

The landscape of the Earth is continuously changing, particularly under a warming climate and widespread human intervention. Consequently, there is a growing demand for monitoring programs focusing on understanding the causes and consequences of landscape change across the globe. One of the key tools that has been employed to track

landscape change is remote sensing, for its ability to perform repeated observation over large geographic areas at a relatively low cost [1–3].

Recent advancement in structure-from-motion (SfM) technology has significantly improved our ability to reconstruct the 3D structure of the Earth's surface, which is further aided by the wide availability of SfM processing software and cameras that are used for data collection [4,5]. This technology is especially useful for monitoring and modeling landscape change in three-dimensional space (e.g., topographic change). Furthermore, SfM has commonly been combined with terrestrial time lapse imagery (usually obtained from semi-permanently installed cameras on fixed platforms), to quantify topographic change in a time series with a view to enhance our ability for continuous and long-term monitoring of landscape change e.g., [6–10]. However, such monitoring involves acquiring and processing a significant volume of data, stimulating the need for automation. While standard SfM workflows for processing one-time datasets are more or less automated, continuous monitoring of topographic change requires a workflow that is more efficient and streamlined than the standard SfM procedure.

Of particular interest is the elimination of ground control surveys from the workflow, as ground control limits the application of SfM for several reasons: targets for ground control may be difficult to place and survey in remote or dangerous terrain, manual identification of targets in imagery is time consuming, and targets can be challenging to locate in imagery e.g., [8]. In dynamic landscapes, such as those of interest in change detection studies, placement of long-term control points may be infeasible. This difficulty is especially important in glaciological studies where the landscape is both continuously moving and melting. Finally, SfM is increasingly used with historical imagery where control points are not available at all e.g., [11,12].

Despite these drawbacks, many studies continue to use manually located control points for scaling and georeferencing SfM surface reconstructions due to lack of other reliable and easy to use options e.g., [6,8,13]. For example, Mallalieu et al. [8] manually identified control points in 18 scenes of a glacier margin, which were selected out of a set of over 1200 images from 426 days of camera operation. Similarly, James and Robson [6] used five control points to reference 37 image pairs of an active lava flow. Other studies have relied on automatically detected control points through pattern recognition [7,14] or through extraction of control points from reference data [10], a technique which is useful where ground control can be placed, but still limits the range of applications where terrestrial imagery can be used for monitoring. Without control points, SfM reconstructions exist in a relative coordinate space rather than a real-world space.

To address the challenges of using control points, several studies have used referenced data from a secondary source (such as a georeferenced point cloud) to georeference unreferenced data [15–18]. For example, Makadia et al. [15] used an extended Gaussian image to describe two point clouds as a whole and calculate the optimal rotation and translation to align them and apply georeferencing from one to the other. Similarly, Bernreiter et al. [18] employed a comparison in the Fourier domain that allows for calculation of the translation and rotation for optimal alignment. These approaches make use of full-cloud characteristics, making them computationally efficient, but are not designed to deal with differences in scale between two clouds. In situations where point clouds are derived from different sources, it is critical to be able to account for this scaling difference as relative coordinate systems may differ significantly.

Other approaches calculate and match descriptors of unique features on each cloud (keypoints). The most common of these is the Fast Point Feature Histogram [19], which relies on a user-specified radius to describe the surface around each point. By computing descriptors at different resolutions, the most unique features can be used for aligning two clouds through a random sample consensus approach, which rejects outliers and minimizes an error function to calculate a best fit transformation between a source and target point cloud. This method accounts for different scaling between clouds but is sensitive to the choice of the radius for describing keypoints, which may be dramatically

different when scales are varied between clouds [17]. To address this, Persad and Armenakis [17] developed a method of identifying scale-invariant keypoints, and a robust descriptor that is invariant with scale, rotation, and translation. The authors successfully used their method to register point clouds that are separated by a significant transformation; however, the method is not robust when point clouds have different point densities. Avidar et al. [16] used yet another approach, by describing the viewpoint of a terrestrial point cloud, and identifying candidate viewpoints in a reference point cloud. A subset of the reference cloud is created that represents the viewpoint at each candidate point, and this is then compared to the terrestrial point cloud to find a best fit. The approach is computationally intensive if the candidate viewpoints are not constrained, which the authors achieve by identifying ground points in an urban scene.

While the methods described above are adequate for georeferencing in some cases, there are situations where they are no longer applicable. These circumstances include scenes where placing and surveying control points is infeasible, or where identifying control points in imagery is impractical or impossible; and where reference data are available but differ from the time series data in important characteristics, such as point density, scale and extent. Given the drawbacks of applying existing point cloud registration methods in some circumstances, an alternative approach to georeferencing which can address those scenarios is desirable.

Discrete Global Grid Systems (DGGS) are an alternative way of integrating and processing geospatial data in a globally referenced, hierarchical grid [20]. Each resolution of a DGGS consists of a set of non-overlapping grid cells that completely cover the Earth's surface; cells are easily refined into child-cells; grid-traversal operations and queries are efficient; and each cell has a unique identifier. As a result, DGGS provide a set of grids on which geospatial data can be flexibly represented and processed in a globally referenced, hierarchical manner [21]. In addition, the fixed locations of DGGS cells would aid in tracking changes over time in a continuous monitoring scenario.

Utilizing a DGGS in the georeferencing process affords us many benefits. First, it allows for the use of easily obtained information to roughly place an unreferenced cloud into a georeferenced frame. Second, it provides a grid on which we build a discrete, georeferenced, approximating point cloud representation. Third, it provides an efficient operation to refine our grid-based representation, which is exploited in our hierarchical optimization algorithm to find a coarse registration. Fourth, and finally, the discrete approximation of the point cloud helps to overcome differences in density between the clouds; for example, by placing the points from both point clouds into grid cells representing the same georeferenced areas, we can calculate corresponding density-invariant descriptors for each cell such as an approximate position of the points as well as an approximate planar orientation.

Based on the strengths of DGGS, in this paper we introduce a new approach to referencing an unreferenced point cloud (UPC) using a georeferenced point cloud (RPC), which (i) addresses differences between densities, viewpoints, and extents of the two clouds, (ii) requires minimal and easy to obtain user input, and (iii) is flexible enough for many applications (including natural landscapes and manmade environments).

## 2. Algorithm Background

Terminology related to the coregistration of point clouds and images varies widely in the literature. In this manuscript we will use the terms coarse and fine registration to refer to the initial rough alignment of two point clouds and the subsequent refinement of that initial alignment, respectively. The terms global and local registration are also often used in the literature with varying meaning. Here, we define the global approaches as emphasizing the importance of the whole scene, while local approaches emphasize the importance of specific locations in the dataset.

To assess the methods described below, we discuss the accuracy, precision, and uncertainty of data and outcomes. We use the term accuracy to mean the proximity of a location

to its true coordinates and precision as a measure of the repeatability of a measurement. The uncertainty of a measurement is the ambiguity stemming from instruments, pixel sizes, and data noise.

### 2.1. Conditions for Applying the Algorithm

The algorithm we describe below relies on the availability of key information about the scene location and camera setup. First is a referenced dataset (RPC) such as a point cloud or digital elevation model that contains the unreferenced dataset (UPC). The flexibility to use an RPC derived from a variety of methods (aerial surveys, satellite data, or terrestrial laser scanner) means that little effort is necessary to derive these data (as compared to the effort of surveying and identifying control points in time lapse sequences). In addition to the RPC, the camera locations of the terrestrial survey are required in two reference frames. The first is the reference frame of the RPC (such as latitude and longitude or UTM), and the second is the relative reference frame of the UPC. Positions in the RPC frame can be rough (e.g. positions from handheld GPS, $\pm 10$ m). Positions in the UPC frame can be obtained from the SfM processing and must be precise in relation to the UPC points.

Several assumptions about the data must also be met to use the approach we describe below. The first is that the RPC represents the true surface, meaning it is correctly georeferenced and undistorted. Second, the UPC is internally consistent, so that when scaled appropriately, point to point distances coincide with true distances between points on the ground. This internal consistency is a function of the SfM processing. The third assumption is that there is little to no change between the RPC and UPC; in dynamic landscapes, the two datasets should be coincident in time, however, in more stable landscapes the timing may be less critical. Finally, the UPC must be contained within the RPC so that the topography of the UPC is represented in a portion of the RPC, although the extents do not need to match.

### 2.2. Representing Point Clouds in a DGGS

DGGS provide a hierarchical set of globally georeferenced, non-overlapping cells (our chosen DGGS uses triangle shaped cells) [22]. As discussed above, utilizing a DGGS in the georeferencing process affords us many benefits for our registration algorithm, which we repeat here. First, it allows for the use of easily obtained information to roughly place an unreferenced cloud into a georeferenced frame. Second, it provides a grid on which we build a discrete, georeferenced, approximating point cloud representation with natural correspondence between clouds. Third, it provides an efficient operation to refine our grid-based representation, which is exploited in our hierarchical optimization algorithm to find a coarse registration. Fourth, and finally, the discrete approximation of the point cloud helps to overcome differences in density between the clouds. To utilize this framework, the point clouds must be encoded into a DGGS-based representation.

We will briefly describe the design of our DGGS and its benefits in this paragraph, but for more information the reader is directed to Hall et al. 2022 [22]. Our chosen DGGS utilizes equal-area triangle cells with a 1 to 4 congruent, center-aligned, refinement. Triangles are efficient to process, refine and render and provide a simplex which has vertices that are guaranteed to be co-planar, which makes our planar approximation calculations simpler and efficient. Additionally, a congruent refinement benefits our hierarchical search as the refined cells are fully contained within the parent cells. This aids the hierarchical search algorithm as successive refinements are always sub-areas of the cells which represent the best matches at the current resolution. While we hypothesize that other DGGS may be used with this algorithm to achieve similar results—this has not been tested.

Our DGGS-based representation starts with the geometry of the DGGS cells, but modifies them to approximate the topography of the point cloud. At a high level, the resulting approximation is a set of DGGS cells, where each cell is oriented and translated to match the position and orientation of an approximating plane, which is calculated from the subset of points that are contained within a given cell at a given resolution (Figure 1). Choosing

a suitable resolution is an important aspect of the algorithm. For example, three distinct points are usually enough to define a unique plane, although more may be needed if the points are collinear. Lower resolution cells are larger and can contain more points, which may improve planar approximations of the local surface. However, at significantly lower resolutions, the local approximation may average out topographic features and poorly represent the surface.

Both the UPC and RPC are partitioned into smaller point clouds by grouping points, which are contained within the geographic boundaries of an individual DGGS cell (Figure 1). Points that fall directly on cell boundaries are deterministically assigned to one of the adjacent cells. For each per-cell point cloud, an approximating plane is calculated using the average location of the points and principal component analysis of the point coordinates to estimate the normal of a plane in 3D space [23]. The underlying DGGS cells are then rotated so their normal is aligned with the normal of the approximating plane and translated so their mid-point coincides with the average location of the per-cell point cloud. As the underlying DGGS cells from our chosen DGGS are triangles, the resulting DGGS representation is a set of triangle approximations and normals for each cell. The full set of triangles is a discrete, georeferenced approximation of the entire point cloud, which we call the DGGS Point Cloud. We use a subscript $D$ to denote the DGGS based representation of the RPC and UPC as $RPC_D$ and $UPC_D$, respectively.

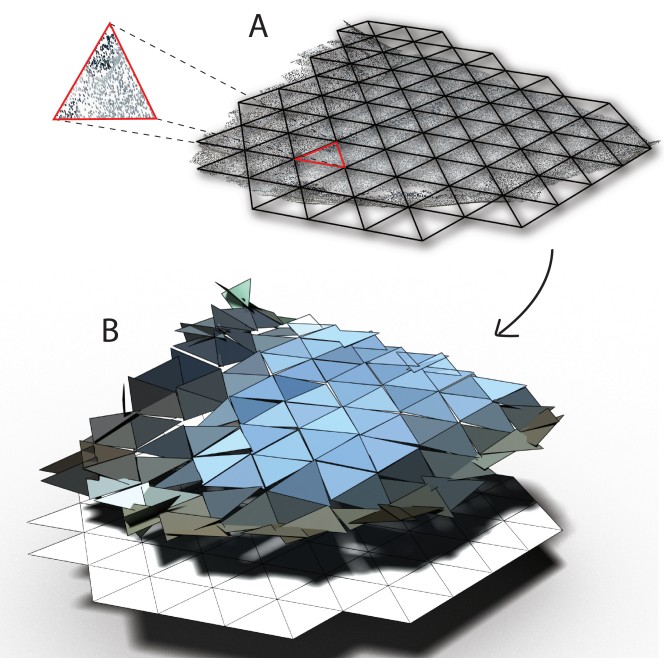

**Figure 1.** (**A**) The point cloud is approximated in the DGGS by partitioning the points into smaller per-cell point clouds based on their geolocation within the DGGS cells. (**B**) A local planar approximation is calculated for each per-cell point cloud, and the DGGS cell from (**A**) is modified by orienting and translating it to coincide with the planar approximation.

### 2.3. Iterative Closest Point Alignment

Iterative closest point (ICP) is a well-known method for the registration of point clouds ([24] at the time of writing, this original manuscript has over 22,500 citations). ICP is an excellent tool for finely registering two point clouds, particularly when the two clouds are already coarsely registered. However, ICP can become stuck in local minima and thus is not useful when coarse registration is needed. In the algorithm we present below, we limit ourselves to the challenge of finding an initial coarse registration and then exploit the strengths of the ICP algorithm to perform the fine registration. In addition, we take motivation from the ICP algorithm and design our coarse registration process in a similar fashion.

ICP iteratively minimizes the root mean squared error (RMSE) of distances between corresponding points in two clouds. Point correspondence is calculated by finding nearest neighbors and in each iteration a transformation is found which minimizes the RMSE between the corresponding points. When using ICP for large point clouds, a randomly selected subset of points is used in each iteration until the RMSE satisfies a given threshold or a certain number of iterations have passed, depending on the application. We make use of the ICP algorithm for our fine registration. To support this we also use one step of ICP between the $RPC_D$ and $UPC_D$ during the coarse registration. In the multi-resolution optimization described below, a single best-fit transformation is calculated at increasingly finer resolutions. Like ICP, we use a correspondence between points, and RMSE of paired distances as the error function being minimized in our algorithm. Unlike ICP, we calculate this distance between the triangle approximations of cells which have the same ID within the $RPC_D$ and $UPC_D$, rather than nearest neighbors (Algorithms 1 and 2). This is a natural correspondence which exploits both the referenced nature of the DGGS grid and the discrete point cloud representation within it. Additionally, the discrete, approximating nature of our representation is robust against density variations within a single point cloud or between the UPC and RPC.

---

**Algorithm 1** Assume the ordering of the incoming lists represents the correspondence, and find a single best transformation to bring the candidate cells as close to the RPC cells as possible. Measure the RMSE between points. This is the cost function/score for a given candidate solution.

---

**Precondition:** The $i$th triangle in each list represents the same cell
**Precondition:** Each vertex of the paired triangles also represent the same vertex in the triangle. This is typically provided by the DGGS

1: **function** MODIFIEDICP($\dot{u}_a \in [\bar{C}_c], \dot{r}_a \in [\bar{C}_{\mathbf{RPC}}]$)          ▷ Candidate and RPC triangles

2:     **moving** $\leftarrow \begin{bmatrix} — & \dot{u}_{a,0,0}^T & — \\ — & \dot{u}_{a,0,1}^T & — \\ — & \dot{u}_{a,0,2}^T & — \\ — & \dot{u}_{a,1,0}^T & — \\ — & \dot{u}_{a,1,1}^T & — \\ — & \dot{u}_{a,1,2}^T & — \\ & \vdots & \\ — & \dot{u}_{a,n,0}^T & — \\ — & \dot{u}_{a,n,1}^T & — \\ — & \dot{u}_{a,n,2}^T & — \end{bmatrix}$          ▷ vertices of candidate solution in $nx3$ matrix

3:     **fixed** $\leftarrow \begin{bmatrix} — & \dot{r}_{a,0,0}^T & — \\ — & \dot{r}_{a,0,1}^T & — \\ — & \dot{r}_{a,0,2}^T & — \\ — & \dot{r}_{a,1,0}^T & — \\ — & \dot{r}_{a,1,1}^T & — \\ — & \dot{r}_{a,1,2}^T & — \\ & \vdots & \\ — & \dot{r}_{a,n,0}^T & — \\ — & \dot{r}_{a,n,1}^T & — \\ — & \dot{r}_{a,n,2}^T & — \end{bmatrix}$          ▷ vertices of **RPC** solution in $nx3$ matrix

4:     $\mathbf{T}_{icp} \leftarrow$ UMEYAMA($moving, fixed$)          ▷ Calculate optimized transform
5:     $transformed \leftarrow []$
6:     **for** $i \leftarrow 0$ to $n$ **do**
7:         $transformed \leftarrow [...transformed, \mathbf{T} \times \bar{C}_{c,i}]$          ▷ Transform triangle points
8:     **return** RMSE($transformed, \bar{C}_{\mathbf{RPC}}$)

---

**Algorithm 2** Calculates the RMSE between two sets of triangles which approximate the point clouds.

---

**Precondition:** Both lists have the same number ($n$) of elements

1: **function** RMSE($c \in [\bar{C}_c], r \in [\bar{C}_{\textbf{RPC}}]$) ▷ Candidate and RPC approximations
2: $\quad r \leftarrow 0$
3: $\quad$ **for** $i \leftarrow 0$ to $n$ **do**
4: $\quad\quad$ **for** $j \leftarrow 0$ to $2$ **do**
5: $\quad\quad\quad r \leftarrow r + \|c_{i,j} - r_{i,j}\|$ ▷ Distance between points
6: $\quad$ **return** $\frac{r}{n}$

---

## 3. Algorithm Description

The georeferencing approach we have developed consists of six stages: initial data input, user-assisted coarse orientation, automated coarse registration, user inspection of solutions, automated fine registration, and georeferencing data outputs.

During the initial data input phase, the user specifies the RPC, UPC, and camera positions in the RPC and UPC reference frames (Figure 2). From these inputs, the system pre-calculates some of the necessary data structures to support an efficient coarse registration. At this stage we have the approximated $RPC_D$ and a look-at-ray is calculated which emanates from one camera location along or parallel to one axis of the UPC reference frame. The nearest point within the UPC to the look-at-ray is identified as the UPC mid-point.

Recall that the reference frame of the UPC is not the same as the reference frame for the RPC. We are looking for a transformation that can place the UPC into the georeferenced RPC reference frame; however, the SfM process does not provide us with enough information to calculate a transform between the two reference frames. Instead, we assume that the UPC coordinates can be directly interpreted as points within the RPC reference frame. This places the UPC into the RPC reference frame, but the points are incorrectly georeferenced. However, it allows us to approach the problem as finding an optimal transformation within the RPC reference frame. To start, an initial transformation is calculated to place the UPC into the RPC extents using the information provided by the user (Figure 2). The scale factor is calculated using the ratio of the distance between camera locations in the RPC reference frame to the distance between the cameras in the UPC frame. Then an initial translation is used to align the camera locations in the UPC and RPC reference frames. A rotation is calculated to ensure that the look-at-ray (in georeferenced coordinates) is "looking" at the mid-point of the center-most cell within the RPC (the look-at-cell). This aligns the mid-point of the UPC with the middle cell of the RPC. The translation is then modified to move the UPC mid-point along the look-at-ray such that it coincides with the look-at-cell's mid-point. Subsequent transformations are calculated in a similar manner, but where the rotation and subsequent translation modification are updated to look at another cell. We call the point cloud resulting from transforming the UPC with this transformation a candidate point cloud ($UPC_C$).

After the initial transformation, users can change the look-at-cell within the RPC to generally orient the initial $UPC_C$ according to their knowledge of the scene (e.g., from camera set up or qualitative inspection of the RPC; Figure 3). A new transformation is calculated based on the selected look-at-cell, which will be the center of the search space for the automated registration. The user then specifies the uncertainty in the selected look-at-cell, which broadens or tightens the search space by only searching within cells that are within this distance from the chosen look-at-cell.

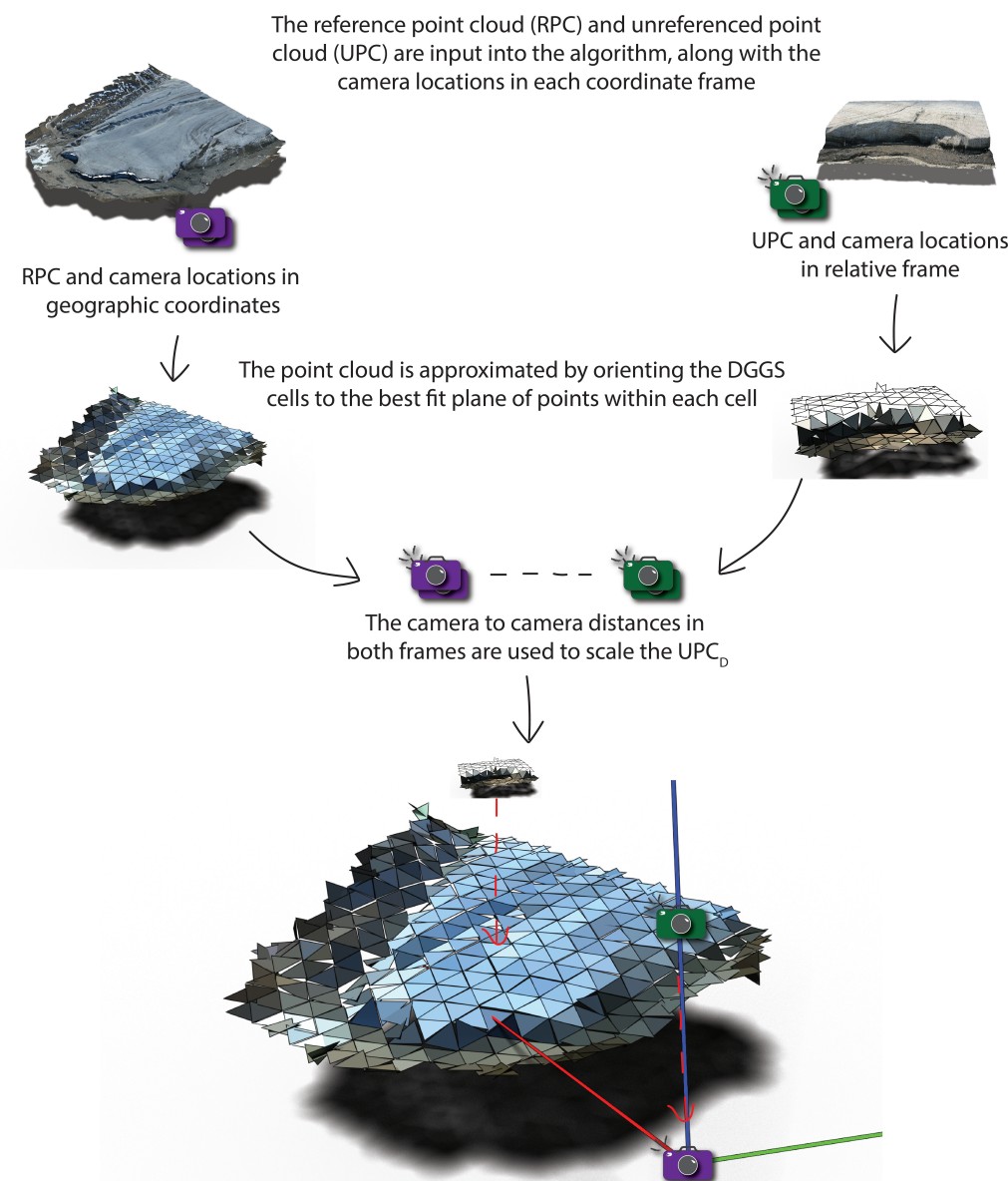

**Figure 2.** Process visualization of data loading into the DGGS for processing and the initial solution calculation before optimization. The reference frame of the UPC is indicated with the red-green-blue axes. The look-at-ray points along the red axis.

The coarse registration is an automated step which uses an iterative, multi-resolution, discrete solution optimization method (Figure 3 and Algorithm 3). The initial search space is set to the cells of interest defined above and proceeds as follows for each cell in the search space:

1. Calculate the transformation as described above and apply it to obtain a new $UPC_C$. Calculate the $UPC_D$ from the $UPC_C$;
2. Perform one step of the modified ICP algorithm and record the RMSE;
3. Repeat for all candidate cells within the specified search space;
4. Sort the $UPC_C$ by their corresponding RMSE and identify the top solutions for further iterations;

5.  Replace the top solutions with their refined children cells (at one level higher resolution), set these as the search space, and iterate.

The user selects a new look-at cell in the RPC$_{D'}$ corresponding to the general location of the UPC$_{D'}$ and adjusts the radius around the new look-at cell to highlight search cells used in coarse registration.

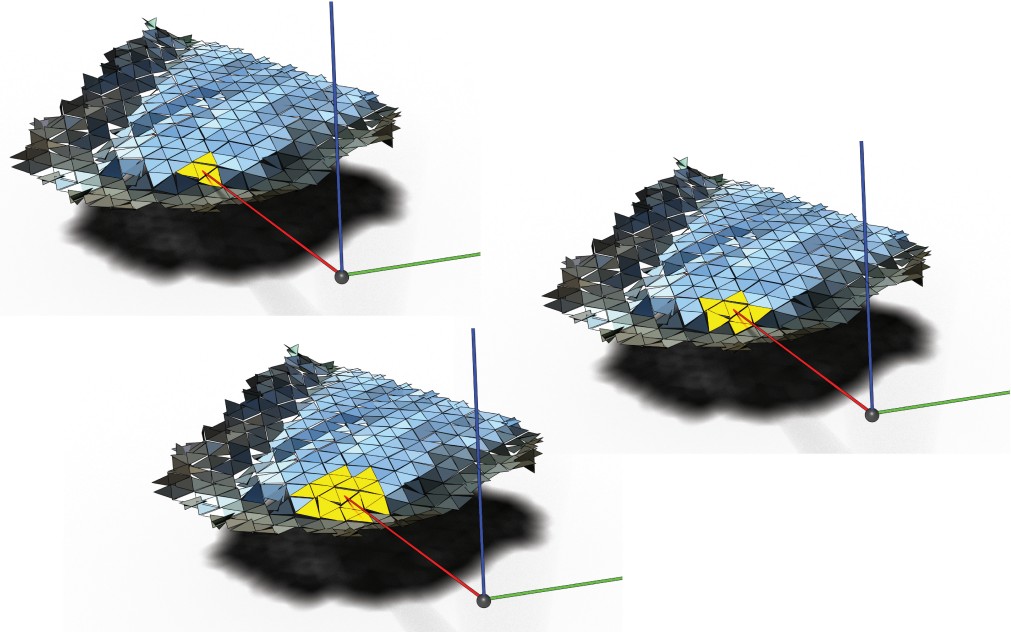

After selecting the search space, coarse registration is performed using an iterative, multi-resolution, discrete solution optimization method

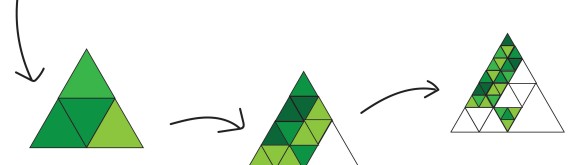

Solutions are sorted based on their RMSE and the cells of the top solutions are split into child cells and the calculation is repeated

**Figure 3.** Process visualization for user selection of search space followed by optimization of fit. The reference frame of the UPC is indicated with the red-green-blue axes. The look-at-ray points along the red axis.

The iteration is stopped when the difference between the best RMSE for the current step is within a sufficiently small threshold, or when the RMSE is larger than that of the previous step, which indicates the resolution is too high for the UPC (i.e., cells no longer approximate the surface well). The number of solutions to keep for each iteration can be adjusted; the default is 50%. Larger percentages of top solutions and larger initial search spaces will increase computation time, while smaller values and smaller search spaces run the risk of missing the best solutions. Setting the initial uncertainty to include all cells within the RPC and keeping 100% of candidate solutions at each step is equivalent to an exhaustive search.

Although the process is automated, the solutions from the final iteration are shown sorted by their RMSE and are displayed along with the RMSE values for the user to explore. Any solution in the top candidates can be selected for fine registration. At this stage, the result of the coarse registration can be saved as a georeferenced point cloud or applied to a set of target point clouds. We make use of this for both case studies in the following

section, which allowed the ICP algorithm to be applied in the CloudCompare software v 2.13 [25]. This fine registration can then be used to transform a set of point clouds (e.g., a time series) and store them as georeferenced point clouds in the RPC coordinate frame. Information on the implementation of the algorithm can be found in Appendix A.

---

**Algorithm 3** Perform a hierarchical search to find the *k* best solutions.

---

1: **function** OPTIMIZE($u \in$ **UPC**, $r \in$ **RPC**, $target \in \ddot{\mathbb{C}}_{\mathbb{I}}$, $d \in \mathbb{R}'$, $k \in \mathbb{Z}$)
2:      $c \leftarrow$ SEARCHAREA($u, r, target, d$)
3:      $best \leftarrow$ FINDBESTCANDIDATES($u, r, c, k$)
4:      **while** KEEPGOING($best$) **do**               ▷ See text for discussion
5:          $c \leftarrow$ REFINE($best$)               ▷ *refine* is provided by the DGGS
6:          $best \leftarrow$ FINDBESTCANDIDATES($u, r, c, k$)
7:      **return** SLICE($best, k$)

---

## 4. Case Studies

In this section, we used the algorithm described above to georeference data from two case studies. Both case studies used imagery from terrestrial time lapse cameras to reconstruct the UPC. The processing for these images is described together, followed by detailed descriptions of both cases.

### 4.1. Data Processing

The UPCs for both case studies described below were produced using the same procedure. Imagery was preprocessed by checking the image quality using the open source Photo4D python package [9]. Image quality was estimated from the image brightness in the EXIF data and blurriness (defined as the variance of the image Laplacian), eliminating images taken at night, or in cloudy conditions. Good quality images were then paired by matching time stamps between cameras, and the pairs were used for SfM processing. In both case studies imagery from two dates was selected to demonstrate the capability of the method, and the ability to apply coarse registration results to multiple time steps.

The SfM reconstruction was carried out in Agisoft Metashape Pro v. 1.5, following the methods of Cook and Dietze [26]. With this method, images from both time steps are used to obtain the camera alignment before splitting the project into individual time steps (Figure 4). This ensures that both time steps share the same camera orientation information and accounts for small changes in the camera parameters over time. After the initial camera alignment using all images, the tie points were filtered to remove points with high uncertainty, camera positions were optimized, and then saved in the relative coordinate frame. The images were then separated into time steps and a dense cloud was reconstructed for each step.

### 4.2. Fountain Glacier

Fountain Glacier is a small glacier located on Bylot Island, Nunavut, Canada (Figure 5A). It is approximately 1 km wide and 16 km long, with two dry calving fronts at the terminus (North and East facing). Melt rates in the summer on Fountain Glacier are 5 cm per day on average [27].

In July 2010, two 10 mega-pixel Canon XTi cameras with 50 mm lenses acquired daily imagery of the glacier terminus at local noon [28]; these cameras were positioned to monitor the north-facing calving front from a moraine, approximately 600 m away (Figure 5A), with convergent view angles and separated by 84 m. The close range of the cameras results in a ground sampling distance (GSD) ranging from 0.05 to 0.08 m across the scene (Figure 5B; Table 1). The convergent and slightly downward view angles, combined with the vertical calving face, led to an internally consistent reconstruction, meaning the reconstructed scene differs from reality only in scale with little to no warping or stretching. Images from 10 and 11 July 2010 were used to produce two point clouds 24 h apart following the methods

described above; the 10 July point cloud served as the UPC. Five on- and off-ice targets were surveyed on 3 July 2010 with a differential GPS, and were used for validation of the georeferencing results. The accuracy of these survey targets is ±0.15 m.

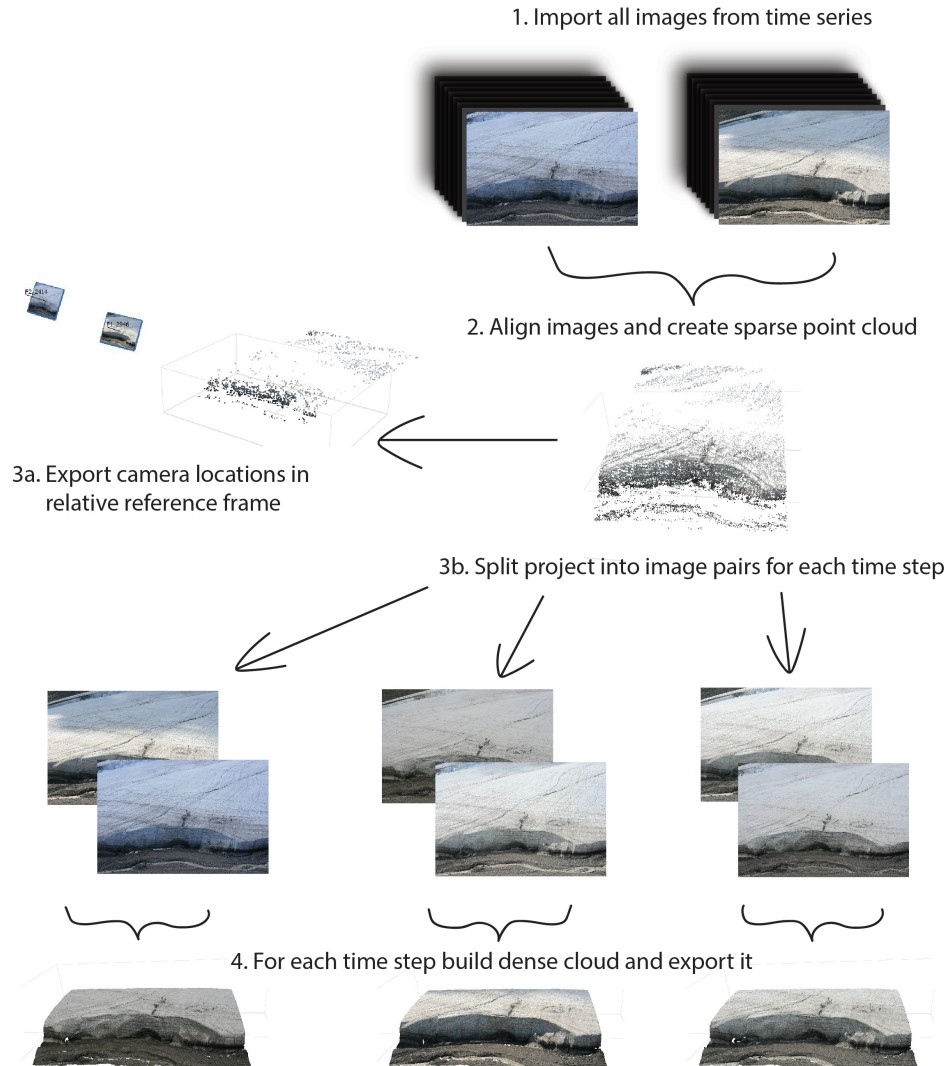

**Figure 4.** Workflow of SfM processing. (**1**) All images from both cameras are input into one Agisoft Metashape project for initial camera alignment and sparse cloud creation (**2**). The camera locations in the relative reference frame are exported ((**3a**) can be performed at any point after 2). The camera alignment and sparse cloud are copied into separate projects for each time step and images from other time steps are removed (**3b**). The dense cloud is built at each time step based on the camera alignment from all images, but the image depths from only that time step (**4**).

A remotely piloted aircraft survey was conducted on 1 July 2010, in conjunction with a survey of control points (also using differential GPS) for SfM processing (survey characteristics are described in [5]). The imagery from this survey was processed in Agisoft Metashape Pro v. 1.5, and control points were manually identified to produce a point cloud, which was used as the RPC. The extent of the RPC was larger than that of the UPC, as seen in Figure 5A.

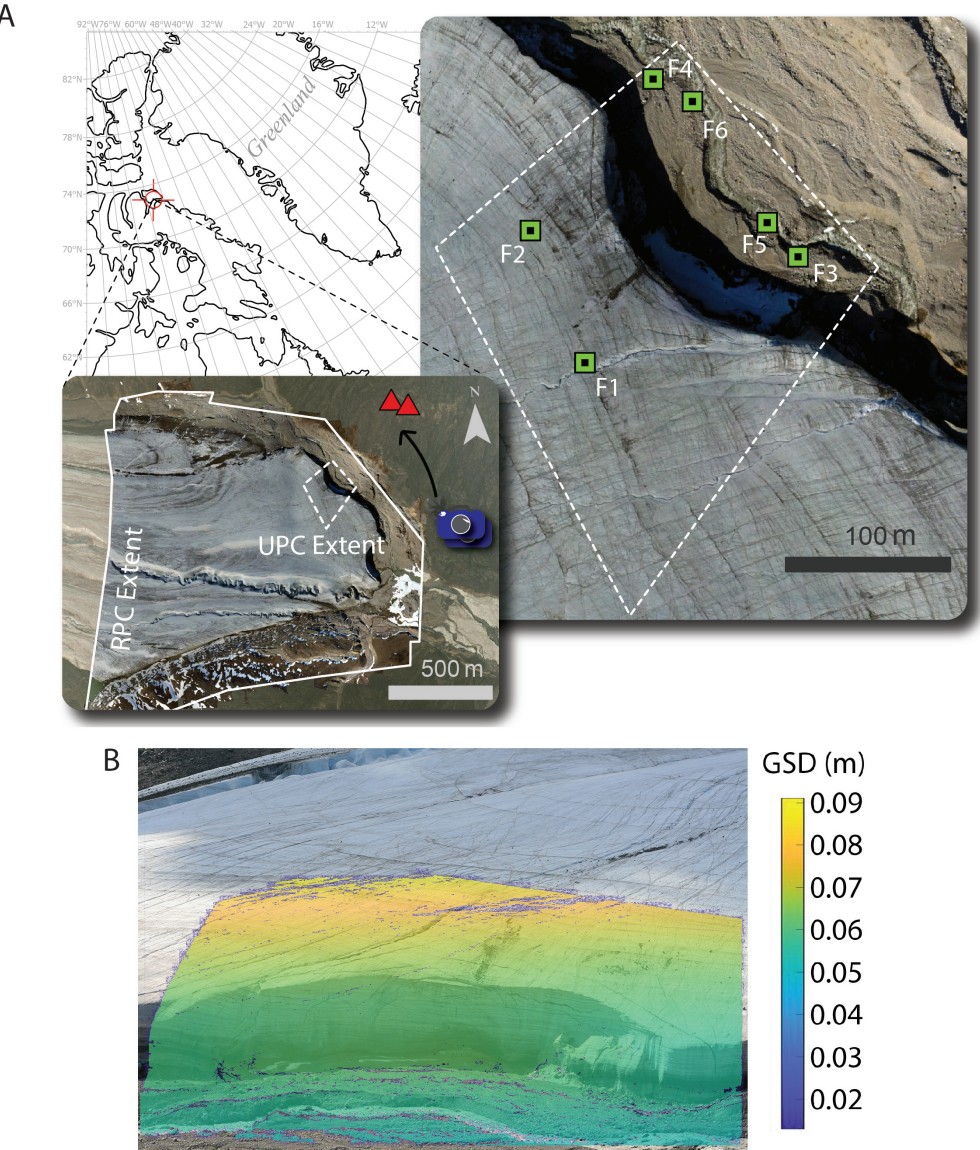

**Figure 5.** (**A**) Fountain Glacier, Nunavut, Canada, was surveyed in July 2010. Two cameras were left in place collecting daily stereo image pairs (red triangles). The extent of the RPC (RPC; solid outline) is shown in relation to the unreferenced point cloud (UPC; dashed outline). Surveyed control points are shown in green. (**B**) This example image from one of the cameras shows the view of the camera with the ground sampling distance (GSD) superimposed.

The alignment algorithm was used, with the RPC and UPC described above, to georeference both the 10 July and 11 July reconstructions. The accuracy of the georeferencing was assessed by calculating the difference between the UPC and RPC. The median difference between the UPC and RPC was 0.065 m, with a variance of 0.29 m (Figure 6A; Table 1). This difference is the same as the GSD of the UPC at the calving front. The variance in the difference can be partially explained by the time difference between the RPC and UPC, during which melt occurred, changing the topography of the ice. An additional accuracy assessment was performed by calculating the difference between the UPC and a point cloud reconstructed in Agisoft Metashape Pro v. 1.5 from the 10 July photos, and georeferenced using the targets during the SfM processing. The median difference between the target-referenced cloud and the UPC was 0.17 m, with a variance of 0.33 m. Given the accuracy of the targets used in georeferencing and the change in target positions between

the survey (3 July) and image acquisition (10 July), the differences are within the positional uncertainty of the referenced cloud.

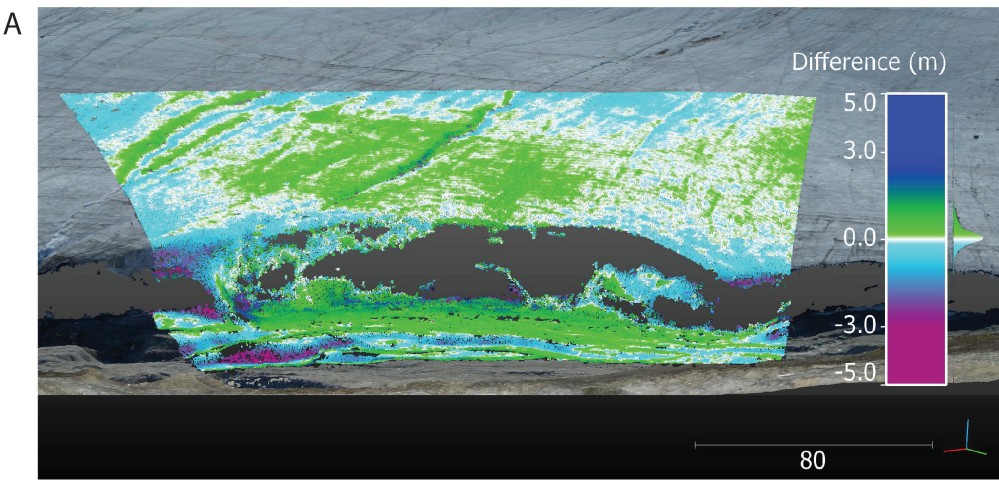

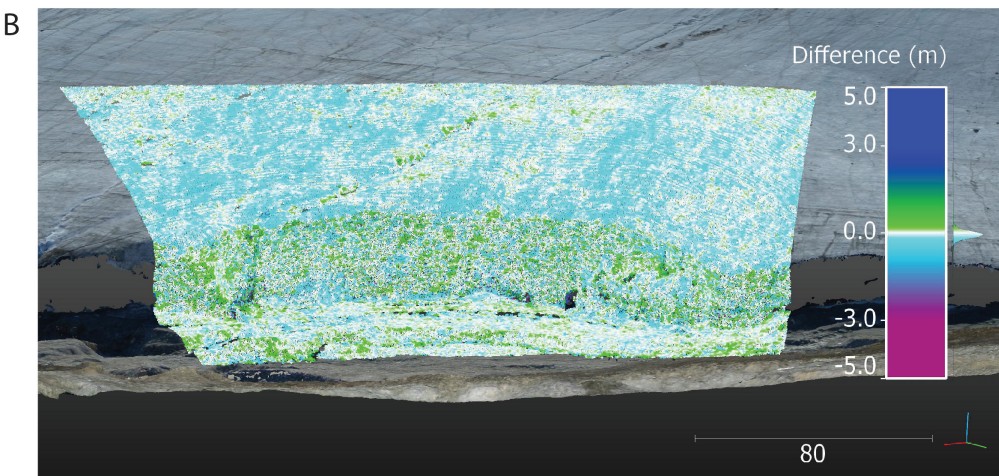

**Figure 6.** (**A**) The difference between the 10 July UPC and the RPC (1 July) for Fountain Glacier, after the UPC has been georeferenced. (**B**) The difference between the 10 July UPC and the 11 July point cloud, after georeferencing. White points in (**A**,**B**) show differences between −0.05 and 0.05 m.

**Table 1.** Characteristics and results from case studies at Fountain Glacier, NU, and Nàlùdäy, YT.

|  | **Fountain Glacier** | **Nàlùdäy** |
|---|---|---|
| GSD (m) | 0.05–0.08 | 0.25–1.4 |
| UPC Date | 10 July 2010 | 25 July 2021 |
| RPC Date | 1 July 2010 | 27 July 2021 |
| GCP Survey Date | 3 July 2010 | 22–23 July 2021 |
| In-situ surface melt (m) | 0.05 | 0.10 |
| GCP Uncertainty (m) | 0.15 | 5 |
| Accuracy RPC-UPC (m) | 0.065 ± 0.29 | 0.25 ± 2.13 |
| Accuracy Referenced-UPC (m) | 0.17 ± 0.33 | 8.19 ± 14.08 |
| Precision on ice (m) | 0.073 ± 0.1 | 0.85 ± 1.7 |

The precision of the georeferencing process was assessed by computing the difference between the 10 July and the 11 July georeferenced point clouds when minimal change was expected. The median difference in the glacier forefield was −0.002 ± 0.09 m, and −0.073 ± 0.10 m over the glacier ice (Figure 6B). The 0.073 m difference on the ice surface is nearly equal to expected melt of 5 cm over the time period and, in combination with

the minimal differences in the forefield, suggests the georeferencing and SfM processing produce clouds that are well aligned with each other.

### 4.3. Nàlùdäy

Nàlùdäy is a surge-type glacier in the St. Elias Mountains, on the traditional territory of the Champagne and Aishihik First Nations in Yukon, Canada (Champagne and Aishihik First Nations, 2021; Figure 7A). Past surges of Nàlùdäy have impacted the indigenous peoples in its vicinity through blockage of the Alsek River and subsequent flooding and ecosystem changes. The Dákwanjè (Southern Tutchone) name roughly translates to "fish stop". The relationship of surges of Nàlùdäy to the river and downstream impacts has led to an ongoing study at the glacier, which began in 2018.

Nàlùdäy is approximately 4 km wide and 34 km long, terminating at two lake calving fronts on either side of a nunatak. In July 2021 two 24 mega-pixel Nikon D5600 cameras, with 18–55 mm variable focal length lenses fixed at 55 mm, were situated on the south side of the glacier, 4 km up-glacier from the terminus and overlooking several measurement sites on the ice (Figure 7A). The two cameras were approximately 850 m apart and 450 m above the glacier surface, with nearly complete overlap of the camera viewshed. On 25 July 2021, several additional photos were taken in the vicinity of each camera during servicing, using the same camera model.

This configuration of cameras results in a highly oblique view across the glacier surface, capturing the valley side on the northern margin 5 km away. The resulting ground sampling distance ranged from 0.25 m in the foreground, to 1.4 m on the opposite slope (Figure 7B; Table 1). This oblique view necessitates more than two images for a good scene reconstruction. Following James and Robson [6], two additional images from July 25 were included in the initial alignment of 24 and 25 July surfaces, and the 25 July reconstruction was used as the UPC. The oblique view of the glacier surface in this case study leads to highly distorted pixels in the far distance, particularly when only two camera viewpoints are used, resulting in the compression of distant space on the relatively flat surface (which appears like "missing space" in the final point cloud, such that the opposite valley side is closer than in reality).

On 22 and 23 July 2021, five on-ice targets were placed for use as validation in the analysis below (Figure 7A). These targets were surveyed using either a Trimble NetR9 (N1, N2, and N4) or R7 (N3 and N5) occupying each target center for a minimum of 60 min, after which the positions were post processed using Natural Resources Canada's Precise Point Positioning service [29]. The resulting positional accuracy is <10 cm horizontally and <15 cm vertically, although the uncertainty in the target locations is higher than the measurement accuracy, due to the larger GSD.

A piloted aerial survey was conducted on 27 July 2021, using a Nikon D850, a 45 mega-pixel camera, with a 24 mm lens. Georeferenced images were used to reconstruct the glacier surface using Agisoft Metashape Pro v. 1.5. The survey set up and data processing are the same as described by Bash et al. [30] and the resulting point cloud was used as the RPC. The extent of the RPC and that of the UPC can be seen in Figure 7A.

Like the Fountain Glacier case study, the alignment algorithm was used to georeference both the 24 July and 25 July reconstructions. The accuracy of the georeferencing was assessed by calculating the difference between the UPC and RPC, as well as the difference between the UPC and a point cloud georeferenced during SfM processing with the five survey targets. The median difference between the UPC and RPC was 0.25 m, with a variance of 2.13 m (Figure 8A; Table 1). Like the previous case study, the median difference is similar to the GSD in the foreground of the scene. Between 25 July and 27 July, an on-ice GPS station near N1 recorded a horizontal displacement of 1.80 m, and a vertical displacement of −0.21 m. An ablation pole drilled into the ice nearby showed 0.10 m of melt.

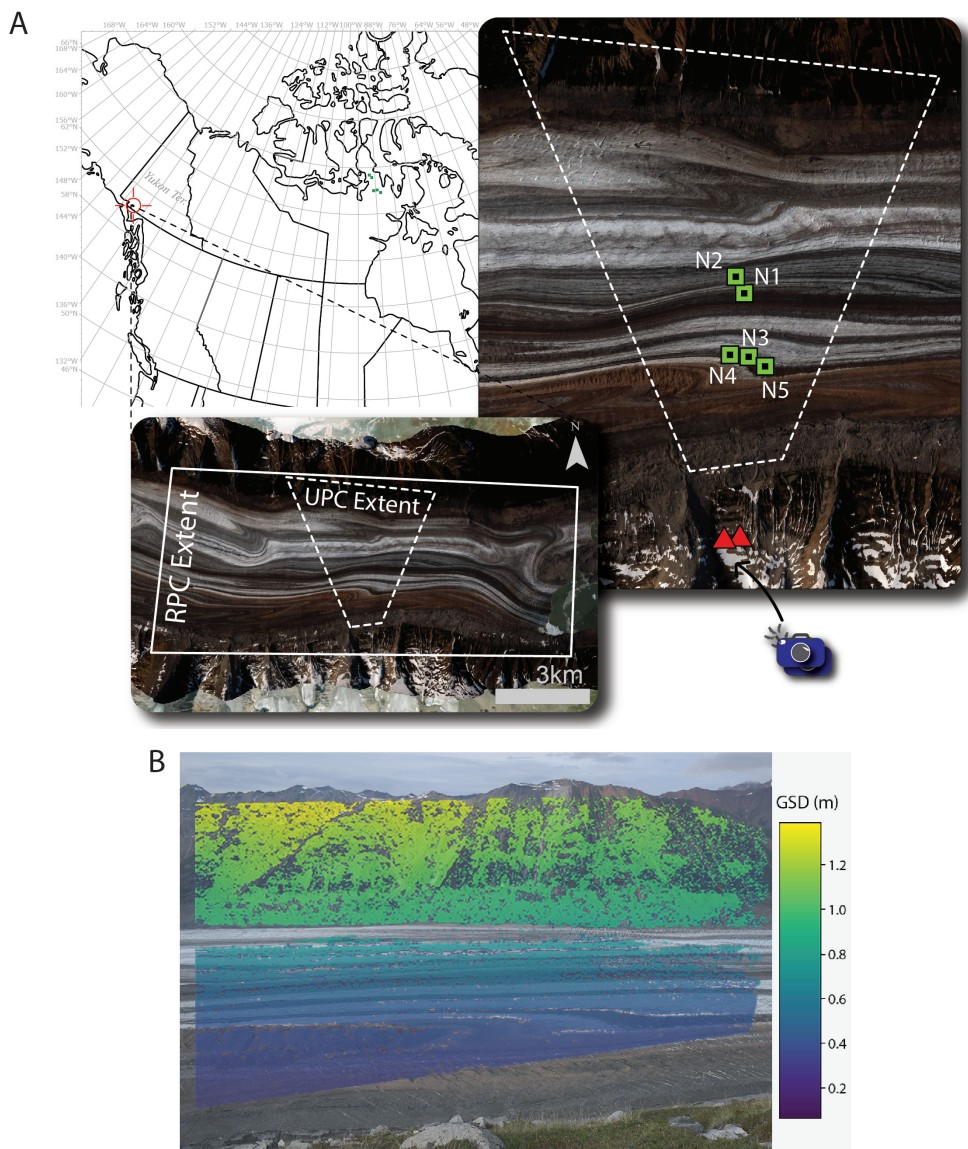

**Figure 7. (A)** Nàlùdäy, Yukon, Canada, was surveyed in July 2021. Two cameras were left in place collecting daily stereo image pairs (red triangles). The extent of the reference point cloud (RPC; solid outline) is shown in relation to the unreferenced point cloud (UPC; dashed outline). Surveyed control points are shown in green. **(B)** Example image from one of the cameras showing the view of the camera with the ground sampling distance superimposed.

An additional accuracy assessment was performed by calculating the difference between the UPC and a point cloud reconstructed in Agisoft Metashape Pro v. 1.5 from the 25 July photos, and georeferenced using the targets during the SfM processing. The median difference between the target-referenced cloud and the UPC was 8.19 m, with a variance of 14.08 m. The location of the targets in the imagery is less certain in this case study than the previous, due to larger GSD. As such, these have a large uncertainty ($\pm 5$ m) and we believe the comparison to the RPC is a more realistic assessment of the georeferencing accuracy. A further comparison of the RPC to the target-referenced cloud showed a median difference of 8.22 m, with a variance of 17.54 m, indicating that the uncertainty in the on-ice targets is the source of the larger discrepancy than was seen in the first case study.

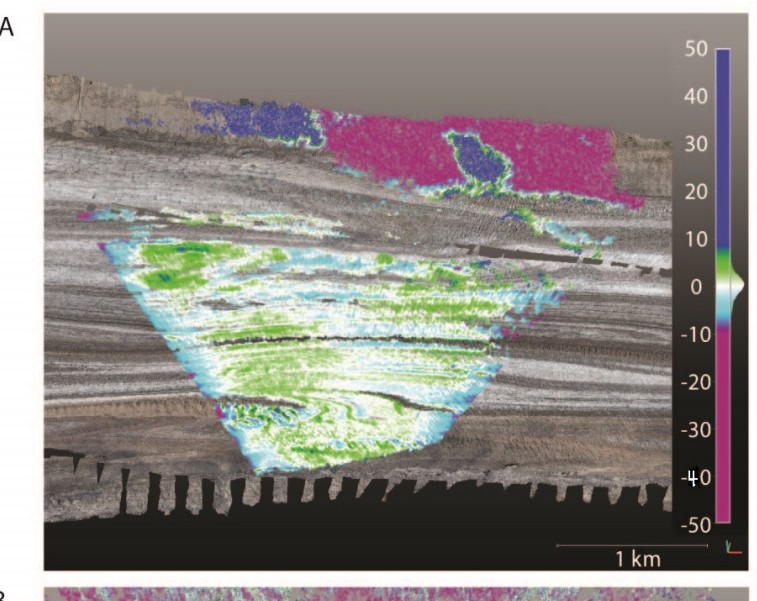

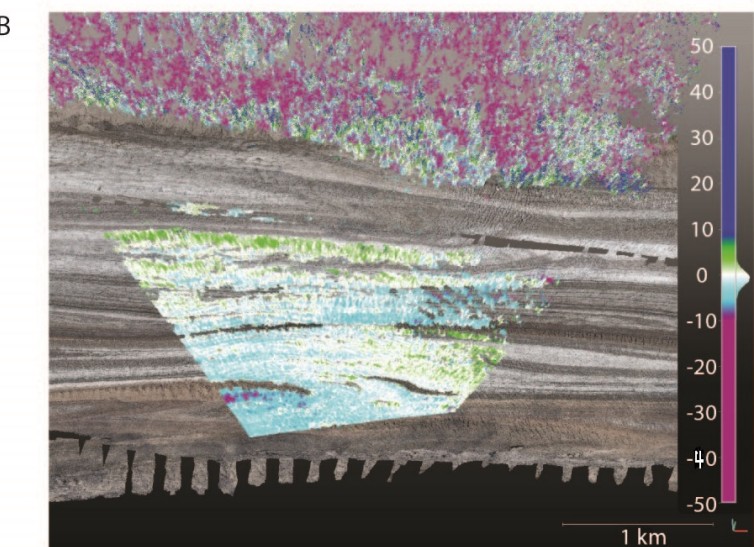

**Figure 8.** (**A**) The difference between the UPC and the RPC (27 July) for Nàłùdäy, after the UPC has been georeferenced. (**B**) The difference between the 24 July point cloud and the 25 July UPC, after georeferencing. White points in (**A,D**) show differences between −1–1 m.

The precision of the georeferencing process was assessed by computing the difference between the 24 July and the 25 July georeferenced point clouds. The median difference over the entire scene was $-1.802 \pm 6.3$ m. The median difference over the glacier ice was $-0.85 \pm 1.7$ m (Figure 8B). During the same period 0.10 m of melt was measured at the ablation pole and displacement at the GPS station was 0.95 m horizontally and −0.10 m vertically.

## 5. Discussion

We present a new approach for georeferencing point clouds that alleviates the need for ground control points for orienting SfM data, while maintaining accuracy within the uncertainty of the data and computational speed. This approach can be applied in many scenarios where remote monitoring is desirable and surveying is not feasible. The minimal interaction of the user with the georeferencing process saves significant time when compared to traditional approaches involving identifying control points in imagery and reference data. In addition, by building on the widely used ICP algorithm within the DGGS, we used a well-tested process for fine registration to achieve the coarse registration

of point clouds. The DGGS also makes the method robust to variations in point density, camera viewpoints and cloud extents, which hinder some previous approaches.

When used to georeference point clouds from two scenes with differing characteristics, this new approach coarsely registered point clouds for further fine registration using ICP. In both case studies differences between surfaces from successive days were similar to expected change over the same time frame. At Fountain Glacier 24-h differences in the forefield were negligible, while on-ice differences were similar to daily melt rates. The median difference at Nàlùdäy between successive days was higher than the measured melt over the time period. Combined with higher uncertainty in the point clouds (due to larger GSD) and flow rates at the site, however, this difference is well within the uncertainty of the measurement. The consistency of both datasets suggests first of all that the point clouds are located in the same coordinate system, but more importantly that the scaling of the unreferenced data is properly solved. If this were not the case, we would expect day-to-day differences to deviate more from expected surface changes.

The accuracy of the final georeferencing is highly influenced by the uncertainty of the input data. The uncertainty stems from the GSD of the original imagery, the uncertainty of GPS positions in the RPC, and the time differences between capturing the RPC and the UPC. In the Fountain Glacier case study, where the GSD was sub-decimeter, the final georeferenced point cloud was within 0.065 m of the reference data. In the Nàlùdäy case study, where GSD ranged from 0.30 to 1 m over the ice surface, the difference between the reference and control data was greater (0.25 m), with a much larger variance. These differences were greater when compared to point clouds georeferenced with survey points, suggesting the algorithm performs well at matching the RPC, but the combined uncertainty in the RPC and UPC leads to lower absolute accuracy in the position of the cloud after georeferencing.

The dynamics of the glacier also plays an important role in the assessment of both the precision and accuracy of the method. The faster flow and greater melt on Nàlùdäy add a minimum of $\pm 1$ m uncertainty to the measurements of accuracy described above, due to the time differences between the RPC acquisition, the UPC acquisition, and the target surveys. Fountain Glacier has much lower flow and melt rates, leading to additional uncertainties of $\pm 0.5$ m. Given the uncertainties, we believe that a comparison of georeferenced point clouds to external data sources should be applied with caution. However, the final georeferencing of these point clouds was highly precise (differences between time steps). This indicates that change calculations between time steps will be reliable; future work will investigate this further.

Using a DGGS was a key component that facilitated the development of our novel georegistration algorithm. A DGGS provides an easy-to-use and efficient hierarchy of global grids, which allows us to consider the problem from a discrete, georeferenced perspective and allows us to use easily obtained information to quickly produce coarse candidate solutions. The discrete nature of the grid also robustly deals with the variations in density within our point clouds. The hierarchy provides an easy-to-use multi-resolution platform and the refinement operations allow us to treat the optimization algorithm as a hierarchical search space of distinct regions, akin to tree-based indexing structures. In addition, a DGGS provides an easy way to model uncertainty in a manner which better represents the discrete, uncertain nature of geospatial data. Finally, the georeferenced nature of the cells used within this point-cloud representation provides a natural correspondence for our modified-ICP algorithm, and when combined with the hierarchical search, it avoids being trapped in local minima that is common in traditional ICP based algorithms.

The method relies on several assumptions, which, if not met, will produce unreliable results:

- The primary assumption is that the RPC and UPC represent the same surface. We have achieved this by choosing an RPC that is contemporaneous with the UPC in one case and nearly so in the other. If the acquisition time of the two datasets differs too significantly, intervening topographic change will hinder the registration process.

The acceptable time difference will depend on the magnitude and rate of change in the study, and future work will seek to quantify how much topographic change is necessary before the registration process fails;

- The SfM processing described in previous sections ensures that a set of point clouds reconstructed from time series are referenced to the same origin. If the time series point clouds are reconstructed independently, they will not share the same camera coordinates in the UPC frame. Without this shared information, the transformation acquired from this method cannot be applied to the whole set;

- The algorithm we present here deals with already reconstructed point clouds, rather than the reconstruction process itself. This necessitates the assumption (as mentioned previously) that the UPC is internally consistent so that, when scaled appropriately, point to point distances coincide with true distances between points on the ground. The consistency of the reconstruction is highly dependent on the scene geometry (i.e., the camera placement and topography of the surveyed area). As we found in the case of Nàłùdäy, two cameras were insufficient to reconstruct the scene.

Scene geometry is affected by the camera placement and orientation, as well as the topography being surveyed; these must be considered to produce data that can be used in this method. Filhol et al. [9] recommended selecting a study site with a concave shape and well-distributed fixed landmarks to ensure high-quality matching between images. We found that in these two case studies the topography of the scene also played a role in the reconstruction quality, specifically the obliquity of the camera view. In scenes where the cameras have a shallow angle over relatively flat topography the GSD becomes distorted with increasing distance from the camera (i.e., the horizontal distances on the ground become stretched into parallelograms). This distortion ultimately leads to compression of space in the far distance in the final reconstruction. We found that, in the reconstruction of Nàłùdäy, two cameras were insufficient to resolve this compression in the final reconstruction, while in the Fountain Glacier case the steep topography of the calving face did not produce the same issue.

In addition, Filhol et al. [9] recommended that the angle between the two camera axes be between 10–20°. This angle depends on the baseline between images and the distance to the area of interest; as the angle becomes smaller the likelihood of indistinct depth reconstruction in the SfM process increases [31]. As the angle increases, the viewpoints begin to differ significantly, and image matching degrades.

In the Fountain Glacier case study, the convergence angle of the cameras was 8°, while in the Nàłùdäy case study the angle was 4°. Giacomini et al. [32] noted that strong correlation is expected between the internal orientation parameters of the cameras (focal length, etc.) when there are fewer cameras in the network. In combination with the convergence angle, this correlation likely contributed to the poor reconstruction at Nàłùdäy in initial tests with only two cameras. Some studies address the issue of solving for the internal orientation parameters by calibrating cameras and specifying the internal parameters e.g., [32]; however, Whitehead et al. [28] found that there was a strong correlation between focal length and temperature, making this approach infeasible for a long time series. This focal length variability results in changes in scaling between time steps when each step is independently reconstructed. We believe that the use of all images in the initial alignment step will address some aspects of the focal length drift, but future work should examine this specifically.

Work on this algorithm is ongoing and we foresee several improvements in the future, in addition to those mentioned above. Currently, we rely on camera positions to calculate the scale factor between the two clouds, but plan to include scale in the optimization process in the future. Early tests in this area resulted in the UPC being scaled to fit in a single cell and reasonable limits on scaling need to be developed before implementing this in the algorithm, for instance by penalizing scales that are dramatically different from the initial calculation based on camera positions.

Both case studies presented above demonstrate this method with complete overlap between the UPC and RPC (that is the UPC is contained within the RPC). In its current form, the algorithm will fail if the UPC only partially overlaps the RPC. Additionally, in the case studies above, the extent of the UPC is 15–20% of the RPC extent. With even closer extents (where the UPC is 20–100% of the RPC), the algorithm should continue to succeed in registering the cloud. We assume that as the percentage becomes smaller (where the UPC is 0–20% of the RPC), the matching quality will degrade and eventually fail.

Another area for improvement is the treatment of gaps in the UPC. Particularly with oblique imagery, topographic occlusion results in gaps and if these are large enough, they overwhelm the RMSE at higher levels of the hierarchical calculation (when cells contain few or no points). A future implementation of the algorithm can address this by using a water-tight mesh representation of the surface, where voids are filled with a linear approximation that connects either side of the gap by using a context-aware void filling such as performed in Wecker et al. [33].

We have presented two case studies to demonstrate this new approach; however, additional test cases will allow investigation of this method to other regions and landscapes. These cases should include tests for some of the preceding considerations, such as quantifying differences between UPC and RPC to determine the necessary similarity, testing the relationship between successful registration and the ratio of the extent between the UPC and RPC, and cases with more comprehensive ground control points for verification. In addition, we hope to test scenarios with varying scene geometry to constrain the relationship between camera placement, number of cameras, and the subject of interest.

We have presented two case studies in glaciated terrain; however, this method is a general approach which can be used in many settings, particularly in areas where control point placement is challenging. By integrating this algorithm into the workflow for monitoring landscape change with time lapse imagery, relatively simple and low-cost monitoring will be more readily available. Much of the work we have relied on in this study has come from research on landslides and slope stability monitoring e.g., [6,7,10,14]. Other studies use SfM to monitor snow accumulation and melt [9], vegetation change [34], and in industrial settings to monitor volume change of gravel or tailings piles [35]. The general nature of the algorithm we have presented here will be useful for these and any other application where the necessary reference data can be collected.

## 6. Conclusions

Here, we presented a new approach to point cloud georeferencing, which requires minimal and easy-to-obtain user input, is flexible enough for many applications, and capitalizes on the DGGS strengths. This algorithm builds on well-established methods using ICP alignment, by translating ICP into the context of the DGGS. By moving away from surveying control points for georeferencing SfM data, we show that georeferencing can be accomplished more simply, with site knowledge (i.e., camera locations and area of interest) in conjunction with reference data that can be collected in a variety of ways.

We have demonstrated the application of the new algorithm in two case studies, where the camera geometry and topography of interest are different. In both cases, the method provided accurate and precise georeferencing, seen in the difference with control data and repeated scene reconstructions. In both cases, the scene geometry was an important consideration to ensure that the SfM reconstruction was internally consistent prior to georeferencing.

Further improvements to this process will address limitations of the current version, which will make this easy to implement in an even greater variety of situations. In addition to algorithm improvements, we envision a wide variety of testing to identify the strengths and limitations of the algorithm. We believe the method we have described will be most useful where topographic change (of a variety of kinds) is being monitored using time lapse cameras, particularly in areas where control point placement is challenging, such as on glaciers and in remote mountainous regions.

**Author Contributions:** Conceptualization, methodology, investigation E.A.B., L.W. and M.M.R.; software, L.W.; data collection E.A.B., C.F.D., K.W., B.J.M., D.M. and L.C.; writing—original draft preparation, E.A.B. and L.W.; writing—review and editing, all authors; visualization, E.A.B.; supervision, C.F.D., G.M. and F.F.S. All authors have read and agreed to the published version of the manuscript.

**Funding:** E.A.B. was supported by a Natural Sciences and Engineering Research Council of Canada (NSERC) Postdoctoral Fellowship. We acknowledge additional support from the NSERC Discovery Program for C.F.D., G.M., F.F.S., B.J.M., D.M. and L.C.; and the NSERC Northern Supplement Program for B.J.M., D.M. and L.C. L.W. was supported by an NSERC Doctoral Scholarship. C.F.D. was also supported by the Canada Research Chairs Program. L.C. received support from the University of Ottawa through their University Research Chair program. M.M.R. was supported by Jan Ciborowski through the NSERC Industrial Research Chair Program.

**Institutional Review Board Statement:** Not applicable.

**Informed Consent Statement:** Not applicable.

**Data Availability Statement:** Data can be accessed at https://github.com/ellie-b/GeoShpr.

**Acknowledgments:** We thank the Polar Continental Shelf Program, Kluane Lake Research Station, Parks Canada, and Icefield Discovery for field support, which enabled aerial surveys of the sites. The Yukon portion of this work was undertaken in the traditional territory of the Champagne and Aishihik First Nations, and the authors are grateful for their permission to undertake this research. Permission to use the traditional Dákwangè (Southern Tutchone) toponym for Lowell Glacier, Nàłùdäy, also spelled Nałudi, was provided by Champagne and Aishihik First Nations and the Kluane First Nation. We also acknowledge the members of the GIV Group and Algorithmic Botany at the University of Calgary for their contributions to the Digital Earth System and their technical insight.

**Conflicts of Interest:** The authors declare no conflict of interest.

## Appendix A. Implementation

The software created during this research is available on GitHub at https://github.com/ellie-b/GeoShpr. It is written in C++17, using GLFW, dear imgui UI framework and the GIV Group Digital Earth https://giv.cpsc.ucalgary.ca/project/digital-earth/. The authors would like to acknowledge the many different open source projects and libraries that were used in the creation of this software, including but not limited to GDAL [36], PDAL [37], libigl [38] and Eigen3 [39].

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
