# Peer review of "A Multi-Resolution Approach to Point Cloud Registration without Control Points"

_remotesensing, doi:10.3390/rs15041161_

Round 1

Reviewer 1 Report

The authors propose a semi-automated approach to perform cloud registration without using control points. Not every time it is possible to include such control points on the scene, and this is one of the advantages of the proposed technique. They use one of the base ideas from ICP that is to minimize the alignment function based on correspondences of points and RMSE of paired distances as error function to be minimized. Differently from ICP, the proposed algorithm calculates the distance between triangle approximations of cells with the same ID between RPC and UPC.

The authors provide a supplementary pdf file with additional information, but they use the same numberings for algorithms, which may confuse the reader.

The keywords of the paper are missing.

Two case studies are used to validate the proposed algorithm. In both cases, the approach is detailed step by step.

I believe the authors should make clear the fact that they had some issues with the 3D reconstruction process when using just two cameras on NàÅ‚ùdäy, since it was not enough to resolve the compression of space in the far distance.

At last, I believe the authors should test the proposed algorithm in a controlled environment, as is, a dataset containing ground truth data in order to verify the results more carefully.

Some minor writing errors were found and are listed as follows.

"large geographic area" -> "large geographic areas"

"topographic change" -> "topographic changes"

"scanner), means" -> "scanner) means"

"points is usually" -> "points are usually"

"triangles are also represent" -> "triangles also represents"

"the user can" -> "users can"

"to“fish stop”." -> "to “fish stop”."

"Point Positioning service [? ]." -> missing reference

"Images and image GPS" -> "Images and GPS"

"while o- ice differences" -> ?

"representation provide a natural" -> "representation provides a natural"

"a linear approximations" -> "a linear approximation"

Reviewer 2 Report

in this paper, the authors present a  novel semi-automated approach for georeferencing unreferenced point clouds (UPC) derived from terrestrial overlapping photos to a reference dataset using Discrete Global Grid System

the paper is well written, we appreciate the authors' effort in deploying the proposed methods details and settings

for further quality improvement, a comparison with existing works is appreciated 

is there any well known data source in this field of search

formatting errors

in Tble 1 : NàÅ‚ùdäy??

Reviewer 3 Report

The manuscript "A multi-resolution approach to point cloud registration without control points" provides a new approach to point cloud georeferencing, using ICP alignment and DGGS. The method is useful for the change-detection of mountain glacier and remote mountainous regions. Overall, the manuscript is appropriate for Remote Sensing.

Minor Comments:

Line 25 :   Keywords?

Figure 8.:  Where is B) and C)?

Line 377:  Point Positioning service [? ]?

Line 426:  What is the meaning of "o- ice differences"

Reviewer 4 Report

This paper proposed a novel approach to georeference unreferenced point clouds by using DGGS, which is especially useful in scenarios where ground control points are unacceptable. The research contributes to the point cloud coregistration techniques and demonstrates a new application domain of DGGS. The manuscript is well-structured and well-written. I have a few comments below.

First, in the Introduction, you talked about the challenges of existing methods of point cloud georeferencing. However, there is a gap between these identified challenges and the adoption of a DGGS – why the usage of a DGGS is necessary and how can it benefit these situations? You have some discussion at the beginning of Section 2.2 (I suggest moving this paragraph to the Introduction), but it is still unclear for the audience regarding the reasons, for example, why the discrete approximation can moderate the multi-density issue.

Second, more explanation on the selected DGGS configuration is needed. For example, why a triangular DGGS is adopted? What is the basic configuration (refinement ratio/basic polyhedron/etc.)? This can be added to Section 2.2.

Third, you mentioned the need for continuous monitoring of topographic change, and DGGS cells have registered, fixed locations at a certain resolution, which exactly benefits the continuous monitoring of geographical phenomena. This might be another reason to use such a data frame.

Round 2

Reviewer 1 Report

I'm satisfied with the corrections performed by the authors. Congratulations, I believe the paper can be accepted for publication now.

Please correct this in the final version of the paper:

"with using the GLFW, dear imgui UI"